# Woven Vascular Stent-Grafts with Surface Modification of Silk Fibroin-Based Paclitaxel/Metformin Microspheres

**DOI:** 10.3390/bioengineering10040399

**Published:** 2023-03-23

**Authors:** Mengdi Liang, Fang Li, Yongfeng Wang, Hao Chen, Jingjing Tian, Zeyu Zhao, Karl H. Schneider, Gang Li

**Affiliations:** 1National Engineering Laboratory for Modern Silk, College of Textile and Clothing Engineering, Soochow University, Suzhou 215123, China; 2Jiangsu Advanced Textile Engineering Technology Center, Nantong 226007, China; 3Department of Applied Physics, The Hong Kong Polytechnic University, 11 Yukchoi Rd, Hung Hom, Kowloon, Hong Kong 999077, China; 4Center for Biomedical Research and Translational Surgery, Medical University of Vienna, 1090 Vienna, Austria

**Keywords:** stent-graft, silk fibroin, drug release, paclitaxel, metformin

## Abstract

In-stent restenosis caused by tumor ingrowth increases the risk of secondary surgery for patients with abdominal aortic aneurysms (AAA) because conventional vascular stent grafts suffer from mechanical fatigue, thrombosis, and endothelial hyperplasia. For that, we report a woven vascular stent-graft with robust mechanical properties, biocompatibility, and drug delivery functions to inhibit thrombosis and the growth of AAA. Paclitaxel (PTX)/metformin (MET)-loaded silk fibroin (SF) microspheres were self-assembly synthesized by emulsification-precipitation technology and layer-by-layer coated on the surface of a woven stent via electrostatic bonding. The woven vascular stent-graft before and after coating drug-loaded membranes were characterized and analyzed systematically. The results show that small-sized drug-loaded microspheres increased the specific surface area and promoted the dissolution/release of drugs. The stent-grafts with drug-loaded membranes exhibited a slow drug-release profile more for than 70 h and low water permeability at 158.33 ± 17.56 mL/cm^2^·min. The combination of PTX and MET inhibited the growth of human umbilical vein endothelial cells. Therefore, it was possible to generate dual-drug-loaded woven vascular stent-grafts to achieve the more effective treatment of AAA.

## 1. Introduction

Abdominal aortic aneurysm (AAA) is a fatal vascular surgical disease, which causes damage to structures near the aortic wall, and the mortality for patients with ruptured AAA is as high as 85% [1]. Vascular stents combined with minimally invasive endovascular aneurysm repair (EVAR) tend to be a gold standard for preventing AAA rupture due to minimal complications and quick recovery compared to open surgery repair (OSR) [2,3]. Proximal aortic control remains a huge challenge for the management of patients with ruptured AAA [4]. Conventional vascular stent grafts are based on a metal stent, such as stainless steel and nickel-titanium alloy, supplemented by a highly biocompatible polymeric membrane covering the surface of the stent [5]. The polymeric membrane was used to cover the vascular rupture, proximal, or distal tumor location, thereby isolating the aneurysm, reconstructing the lumen, and gradually reducing blood pressure within the tumor lumen [6]. However, these vascular stents have poor durable performance after implantation due to their structural defects and long-term exposure to the physiological environment of the body [7], which subsequently leads to a series of complications. For example, in-stent restenosis caused by tumor ingrowth still increases the risk of secondary surgery since these vascular stent-grafts suffer from fatigue [8], fracture [9], endoleaks [10,11], thrombosis [12], and endothelial hyperplasia [13] in middle and long-term clinical trials. Good mechanical properties determine long-term patency after stent placement and are an important factor in whether the stent is qualified or not [14]. Fiber-based vascular stents with excellent mechanical stability can be achieved by textile forming technologies with select fabric tissue, warp and weft materials, and density. Hence, it is of great significance to develop a new type of stent-graft with robust mechanical properties, biocompatibility, and drug delivery functions that can inhibit thrombosis and the growth of AAA.

Diabetes medications, such as metformin (MET), can regulate blood sugar and protect the cardiovascular system and have been used to inhibit AAA growth and induce tumor cell apoptosis [15]. Maestrelli et al. developed a calcium alginate microsphere containing MET that avoided the problem of premature leakage of hydrophilic drugs and further controlled their release in the intestine but did not provide the desired sustained release, exhibiting a sudden release within only 30 min. [16]. Paclitaxel (PTX) is an alkaloid with a unique anti-cancer mechanism [17,18] and has effects on a variety of advanced cancers, including AAA. For instance, Zhang et al. prepared a PTX-loaded polypropylene-co-glycolide (PLGA) microsphere with a microporous surface and a porous internal structure, which had good performance in drug delivery and antitumor effects [19]. However, the poor water solubility of PTX was unfavorable for clinical use. Ethanol and polyethoxylated castor oil (Cremophor EL) were added to PTX to improve its water solubility [20]. However, elevated levels of the surfactant Cremophor EL required PTX administration, which caused undesirable side effects, such as hypersensitivity reactions, which were not conducive to long-term treatment [21].

Silk fibroin (SF), derived from the silkworm, is a protein-based biomacromolecule consisting of hydrophobic and hydrophilic blocks [22]. The hydrophobic blocks are rich in Alanine, Glycine, and Serine, which are responsible for generating the crystalline structure of SF by folding into β-sheets, but the hydrophilic region is a short and non-repetitive segment [23]. Recently, SF has been proven to serve as a potential candidate for drug delivery [24,25] because of its highly controllable composition, sequences, structures, architectures, mechanical properties, and multi-functions [26,27,28]. For example, Li et al. embedded magnesium oxide nanoparticles into the SF system to fabricate MgO-SF spheres with excellent antibacterial and anticancer properties that are bioresorbable and traceable and which have great potential for controlled drug delivery and noninvasive bioimaging application [29].

Here, we present a woven vascular stent-graft modified with SF-based paclitaxel/metformin/methanol microspheres (SF-PTX-MET-MT microspheres), which offers a slow-release drug function and inhibits thrombosis and the growth of AAA. SF-PTX-MET-MT microspheres were synthesized by emulsification-precipitation technology and modified on the surface of a woven stent with electrostatic bonding. The surface morphology, thickness, surface hydrophilicity, radial tensile strength, longitudinal tensile strength, water permeability, microstructure, and drug release properties before and after coating were investigated. The safety and effectiveness of the drug in the covered stent were verified by the human umbilical vein endothelial cell line. 

## 2. Materials and Method 

### 2.1. Materials

Degummed silk was purchased from Xiehe Silk Incorporation (Shengzhou, China). Polyethylene glycol (PEG, 10,000 g/mol) and methanol (MT) were purchased from Sinopharm Chemical Reagent Co., Ltd. (Shanghai, China). MET was purchased from Ketong Incorporation (Suzhou, China). PTX was bought from Bailingwei Chemical Technology (Shanghai, China). Lithium bromide was obtained from Aladdin Industrial Corporation (Shanghai, China). Polyallylamine hydrochloride (PAH) was obtained from Jiangsu Argon Krypton-Xenon Material Technology Co., Ltd. (Suzhou, China). Human umbilical vein endothelial cells (HUVECs) were derived from healthy human umbilical vein tissue and purchased from Biotechnology Co., Ltd, Shanghai Enzyme Research Biotechnology Co., Ltd. (Shanghai, China). They were obtained from the American Tissue Culture Collection (ATCC).

### 2.2. Preparation of Drug-Loaded SF Microspheres

SF solution was prepared as reported previously [30]. Briefly, a 10 wt% SF solution, 50 wt% PEG solution, and MT were blended at a volume ratio of 10:10:1 and precipitated for 12 h in the dark at room temperature. Typical SF microspheres were obtained through repeated centrifugation (12,000 rpm, 10 min), ultrasonic dispersion, and freeze drying [31,32]. For the fabrication of drug-loaded SF microspheres, 0.2 g MET was dissolved in the 1 mL SF solution, then 1 mg PTX was dissolved in the 1 mL PEG solution, and together mixed with MT (*v*/*v*/*v* = 10:10:1); after being incubated in the dark at room temperature for 12 h, deionized water was added to a centrifugal wash by uniformly ultrasonic dispersion, which was repeated three times to obtain drug-loaded SF microspheres. Note that microspheres with different combinations of SF, PTX, MET, and MT were prepared (Figure 1A).

The yield of drug-loaded microspheres was obtained by weight change. In detail, the microspheres were incubated in a centrifuge tube, and the weight of the empty centrifuge tube, initial SF, and dried SF microspheres was recorded. The yield of the drug-loaded SF microspheres was calculated according to Equation (1).
(1)$$\mathrm{Yield}\left(\%\right)=\frac{W_{3}-W_{1}}{W_{2}}\times 100\%$$
where *W*_1_ is the weight of the empty centrifuge tube, *W*_2_ is the initial weight of SF, and *W*_3_ is the dry weight of SF plus microspheres.

### 2.3. Preparation of Stent-Graft with Drug-Loaded Membranes

The stent was woven with PET filaments in plain weave, and the stent-graft was coated with drug-loaded membranes and fabricated, as shown in Figure 1B. The stent-graft was immersed in a NaCl solution (0.15 M, pH: 3) for 60 min to impart a negative charge and was successively immersed in a positively charged PAH solution (10 mg/mL, pH: 7.4) for 15 min and the drug-loaded SF microsphere solution for 15 min before washing to remove the unstable microspheres. The immersion and washing of stent-graft in PAH and microsphere solutions were repeated seven times to obtain the stable stent-graft with drug-loaded membranes.

### 2.4. Characterization

The morphology of SF microspheres and their drug-loaded membranes were observed using a Hitachi scanning electron microscopy (SEM). One hundred drug-loaded microspheres were randomly selected from one SEM image, and the average diameters of these microspheres were analyzed using Image J (1.47 v). The specific surface area of microspheres was estimated from these diameters. Fourier transforms infrared (FTIR) measurements of dried drug-loaded microspheres/membranes from 400 to 4000 cm^−1^ were analyzed with a Thermo spectroscopy instrument. The X-ray diffraction (XRD) intensity of the lyophilized powders of drug-loaded microspheres was recorded by a PANalytical X-ray diffractometer. The diffraction angle ranged from 5° to 45°, the X-ray wavelength λ was 1.540, and the scanning speed was 0.04°/s. The thermogravimetric analysis (TGA) of drug-loaded microspheres (5 mg) was carried out by purging using a TGA instrument from the TA company, and the heating rate was performed at 15 °C/min.

### 2.5. Mechanical Properties and Water Contact Performance of Drug-Loaded Membranes 

The thickness, mechanical properties, and water permeability of drug-loaded membranes were tested according to the ISO 7198 2016 standard. The thickness of the vascular stent-grafts with drug-loaded membranes was measured by the fabric thickness gauge (YG(B)141D). For each sample, five different areas were selected for thickness testing, and the average value was recorded. The measuring area was 0.5 cm^2^, and the measuring pressure was 981 Pa.

Tensile strength in the axial/longitudinal direction of drug-loaded membranes was tested by a material testing machine (INSTRON-3365, USA). The length and width of the samples were 40 × 5 mm to ensure a working distance of 20 mm. The tensile strength was measured as shown in Equation (2).
(2)$$F=\frac{2T}{\pi d}$$
where *F* (N/mm) is the tensile strength of the drug-loaded membranes; *T* (N) is the breaking strength of the drug-loaded membranes; *D* (mm) is the inner diameter of the drug-loaded membranes.

A 1 cm^2^ area of each sample was randomly selected for contact angle measurement using an optical contact angle measuring instrument (Kruss DSA 100). Water permeability, i.e., the water flow per unit area of the drug-loaded membranes in unit time, was measured under the hydrostatic pressure of 120 mmHg. The membrane is non-porous if the water flow rate is less than 300 mL/min·cm^2^. Five areas were tested for each stent-graft, and the average contact angle was taken.

### 2.6. Drug Loading Efficiency and in Vitro Drug Release

The drug loading efficiency of MET and PTX in the microspheres system was determined by establishing an absorbance–concentration standard curve. Therefore, MET/PTX was dissolved in a 9.3 mol/L LiBr solution with seven concentration gradient dilutions, and the absorbance at 231/230 nm of seven concentrations was measured using an ultraviolet spectrophotometer (Cary 5000). The linear regression equations of absorbance with drug concentration were Y_MET_ = 0.07744X_abs_ + 0.39623 (R^2^ = 0.999) and Y_PTX_ = 0.03873_Xabs_ − 0.38474 (R^2^ = 0.999). The encapsulation rate (ER) and drug loading rate (DR) of MET/PTX in the drug-loaded are given in Equations (3) and (4).
(3)$$\mathrm{ER}\left(\%\right)=\frac{\mathrm{MET}/\mathrm{PTX}\;\mathrm{content}\;\mathrm{in}\;\mathrm{actual}\;\mathrm{microspheres}}{\mathrm{Theoretical}\;\mathrm{MET}/\mathrm{PTX}\;\mathrm{content}\;\mathrm{in}\;\mathrm{microspheres}}\times 100\%$$
(4)$$\mathrm{DR}\left(\%\right)=\frac{\mathrm{MET}/\mathrm{PTX}\;\mathrm{content}\;\mathrm{in}\;\mathrm{drug}-\mathrm{loaded}\;\mathrm{microspheres}\;}{\mathrm{Weight}\;\mathrm{of}\;\mathrm{drug}-\mathrm{loaded}\;\mathrm{microspheres}\;}\times 100\%$$

The 10 mg/mL drug-loaded microspheres/membranes were soaked in phosphate-buffered saline (PBS) solution, which was placed on a shaker in an oven at 37 °C and oscillated at a frequency of 100 times/min. The released solution of 2 mL was collected at 1, 2, 4, 7, 12, 24, 48, and 72 h intervals, and 2 mL of fresh PBS solution was added after sampling. The UV absorbance wavelength of 233 nm and 227 nm was used for MET and PTX, and the concentration of MET and PTX was calculated according to the absorbance-concentration standard equation Y_MET_ = 0.11389X_abs_ − 0.1797 (R^2^ = 0.999) and Y_PTX_ = 0.06412X_abs_ + 0.35086 (R^2^ = 0.999), respectively. The cumulative release rate of MET/PTX at each time point is as follows:(5)$$\mathrm{Cumulative}\;\mathrm{release}\;\mathrm{rate}\;\left(\%\right)=\frac{\left(C_{1}+C_{2}+\cdots +C_{n-1}\right)\times V}{1W\cdot X}$$
where: *C_n_* is the concentration of the Nth sampling point; *V*_0_ is the volume of the release medium; *V* is the volume of each sampling; *W* is the total mass of the sample; *X* is the drug loading rate (%).

### 2.7. Cell Culture and Proliferation

HUVEC were cultured as a monolayer in a high-sugar medium at 37 °C and 5% CO_2_. The HUVEC were subjected to recovery and progressed down the passage until more than 80% of the cells had aggregated. Punches of sterilized (^60^Co radiation at 18 kGy) drug-loaded membranes were placed in a 24-well plate and seeded with cells. It can be noted that the same-sized samples of drug-loaded stent-grafts were collected using a hole puncher. Each well was inoculated with 1 × 10^4^ cells. A test group of blank samples was compared to membranes coated with PTX, MET, or a combination of PTX/MET. To calculate cell viability and the inhibition rate of drug-loaded stent-grafts, absorbance values were detected using a Bio-Tek microplate reader with the CCK-8 method. The cell viability and inhibition rate are given by Equations (6) and (7).
(6)$$\mathrm{Cell}\;\mathrm{viability}\;=\frac{X_{s}-X_{b}}{X_{c}-X_{b}}\times 100\%$$
(7)$$\mathrm{Inhibition}\;\mathrm{rate}=\frac{X_{c}-X_{s}}{X_{c}-X_{b}}\times 100\%$$
where: *X_b_* is the absorbance of wells with the culture medium and CCK-8 solution, without the cells and drugs. *X_c_* is the absorbance of wells with cells, the culture medium, and a CCK-8 solution. *X_s_* is the absorbance of wells with cells, culture medium, the CCK-8 solution, and drugs. HUVECs were stained with calcein for 20 min after being cultured for 48 h, and then the cell growth status was observed using an inverted fluorescence microscope from Caliper life science.

## 3. Results and Discussion

### 3.1. Morphology of Drug-Loaded SF Microspheres

SF molecules are composed of hydrophilic and hydrophobic fragments. Its secondary structure can be changed under external physical conditions (ultrasound, freezing, etc.) and chemical stimulations (pH value, surfactants, etc.) from random coiling to α-helix or β-sheet [24,33]. Similarly, in this study, the conformation of SF changed from an unstable random coil to a stable α-helix or β-sheet upon the addition of a polar solvent PEG. The phase separation occurred due to the fact that PEG can more easily adsorb water molecules in the dispersed phase into the continuous phase, causing the SF molecules to lose water and gradually expose the hydrophobic region [34]. Under the hydrophobic force, SF gradually wrapped the PTX molecules and MET molecules to aggregate, precipitate, and form drug-loaded SF microspheres (Figure 2A).

SF microspheres with and without loading drugs showed excellent pellet-forming properties, uniform sizes, and flat roundness, and there was no adhesion phenomenon in Figure 2B, which confirmed that SF is an excellent carrier for drug-loading. SF-MT microspheres have few pores and compact sizes compared with SF microspheres (Figure 2B(c)). Because methanol is a polar solvent, it can help the emulsified SF droplets to quickly dehydrate, accelerate the self-assembly speed of SF, and cause the microspheres to have a large particle size, reaching around 1.0~8.0 μm (Figure 2B(c_1_)). Interestingly, once the drugs were added into the SF microspheres system, their pellet-forming changed. SF-PTX-MT microspheres are rough, porous, and small compared with SF-MT microspheres because PTX is a hydrophobic drug, which has an impact on the hydrophilic and hydrophobic environment of the solution and hinders the self-assembly process of SF molecules. The opposite is that MET is a hydrophilic drug [35], which can quickly dissolve in water molecules and affect the dehydration process of SF molecules, causing small particle sizes to be less porous, with poor roundness and a smooth surface. SF-PTX-MET-MT microspheres were added with two drugs at the same time, although the roundness and uniformity of drug-loaded SF microspheres were poorer, and the average diameters of these microspheres were further reduced. Therefore, the addition of either hydrophilic or hydrophobic drugs can affect the hydrophobic interaction between SF molecules and help decrease their average diameters. The particle size of the drug-loaded SF microspheres incorporating the two drugs was around 0.5~4.5 μm; this small-sized microsphere structure can increase the specific surface area and promote the release and dissolution of the drugs.

### 3.2. Yield Analysis of Drug-Loaded SF Microspheres

The yield of drug-loaded SF microspheres has important an economic reference for the cost control and drug loading of drug preparations. As shown in Figure 3A, the yield of pure SF microspheres is 19.17 ± 5.45%, and the yield of SF-MT microspheres increased by 7.48%, reaching 26.65 ± 9.73%, which confirmed that the addition of methanol could accelerate the dehydration self-assembly process of SF molecules. The same results could also be obtained from the group of SF-PTX and SF-PTX-MT, where the yield increased from 18.38 ± 5.63% to 28.35 ± 10.25%. The yield of SF-PTX-MT, SF-MET-MT, and SF-PTX-MET-MT microspheres presented no significant difference as the drugs were added to the SF microspheres, reaching 28.35 ± 10.25%, 24.85 ± 8.17%, and 27.94 ± 12.39%, respectively; the preparation of the SF microspheres by the PEG emulsification-precipitation method is a dynamic process in which the SF molecules were dehydrated and self-assembled to precipitate microspheres [36], while the addition of drugs did not affect the process, so there was no major effect on the yield of microspheres.

### 3.3. Structural Analysis of Drug-Loaded SF Microspheres

Structural analysis can help to understand the formation mechanism of drug-loaded SF microspheres. The absorption characteristic peaks of Silk Ⅰ appear near 1650–1655 cm^−1^, 1540–1555 cm^−1^, 1235 cm^−1^, and 650–670 cm^−1^, as shown in the peak of Amide I, II, III, and V, respectively. The four peaks of Silk Ⅱ were around 1535–1615 cm^−1^, 1525–1541 cm^−1^, 1260 cm^−1^, and 690–700 cm^−1^. It can be seen from Figure 3B that SF microspheres had a characteristic peak at 1653 cm^−1^ and indicated the presence of α-helix and random coil structures in pure SF microspheres, which mainly belonged to the Silk Ⅰ structure. The peak at 1625 cm^−1^ that accompanied a β-sheet structure was found in the SF microspheres with MT, MET, and PTX, indicating that the structure of the drug-loaded SF microspheres changed from an unstable Silk Ⅰ structure to a stable Silk Ⅱ structure. The O-H absorption peak of paclitaxel was 3469 cm^−1^, and the -NH_2_ absorption peak of metformin was at 3292 and 3368 cm^−1^. These results reveal that the drugs were successfully encapsulated in the SF microspheres.

The main diffraction angles of Silk Ⅰ were 12.2°, 19.7°, 24.7°, 28.2°, and the angles of Silk Ⅱ were 18.9°, 20.4° [37]. As shown in Figure 3C, the main diffraction angles of the pure SF microspheres were 18.9° and 19.7°, which is attributed to the crystalline structure of Silk Ⅱ (crystalline region) and Silk Ⅰ (amorphous region). The main diffraction peaks of drug-loaded SF microspheres were all located at 18.9° and 20.4°, indicating that the addition of MT and drugs induced the transformation of microspheres from Silk I to Silk Ⅱ, which is consistent with the results of FTIR. A stable crystalline β-sheet structure from Silk II of drug-loaded SF microspheres can prolong drug release and prevent membrane degradation.

The TGA results of the drug-loaded SF microspheres are shown in Figure 3D; all six groups of drug-loaded SF microspheres had a mass loss peak at around 100 °C, which resulted from water evaporation. Subsequently, all the sample mass was relatively maintained between 120 °C and 270 °C. In the third stage, all the sample masses decreased rapidly after 270 °C, and this decreasing rate gradually slowed at 450 °C, which was caused by the thermal decomposition of protein. From the results of the mass loss rate in the third stage, the thermal stability of the MT-added microspheres was found to be better than that of the pure SF microspheres due to the MT-added microspheres that had a thermal stable Silk Ⅱ (crystalline area) structure [36]. It was also found that the thermal stabilities of SF microspheres added with PTX and MET were better than pure SF microspheres, which also showed the secondary structure of SF changes from Silk Ⅰ to a stable Silk Ⅱ structure after adding PTX and MET. Structural stability is a fundamental characteristic of stent-grafts [38], and good thermal stability of stent-grafts ensures good performance during storage, implantation, and service [39].

### 3.4. In Vitro Release of Drug-Loaded SF Microspheres/Membranes

The in vitro release of drug-loaded SF microspheres provides information for in vivo behavior after stent implantation. Therefore, the ER, DR, and in vitro release rates were examined. As shown in Figure 4A, the ER and DR of MET in SF-PTX-MET-MT microspheres were significantly better than that of SF-MET-MT microspheres. Based on our measurements, we hypothesized that the hydrophobic force between PTX and SF molecules decreased the ER and DR. However, when MET and SF molecules were combined, the forces appeared to increase, resulting in the increased ER and DR of MET. On the other hand, the ER and DR of PTX in SF-PTX-MET-MT microspheres are significantly better than that of SF-PTX-MT microspheres (Figure 4B) because MET was positively charged and could combine with negatively charged SF molecules through electrostatic attraction.

As shown in Figure 4C, there was a bursting release phenomenon that SF-MET-MT microspheres and SF-PTX-MET-MT microspheres exhibited a fast cumulative release rate of MET within 7 h and remained stable during 24–48 h, reached 31.17 ± 0.89% and 26.65 ± 0.74%, respectively. According to previous findings, this occurred because MET was hydrophilic and applied to the surface of the microspheres through the process of self-assembly, and the dehydration of SF molecules [40]. Particularly, the cumulative release rate of PTX in SF-PTX-MET-MT microspheres reached 42.78%, which is 1.2 and 1.5 times that released by SF-PTX-MT microspheres and SF-PTX microspheres (Figure 4D). This could indicate that MET is easier to combine with PTX because of hydrophilic and hydrophobic attraction, leading to the greater presence of the PTX molecular in the early stage of the sustained releasing process and the final cumulative release rate became higher.

### 3.5. Performance on Stent-Graft with Drug-Loaded Membranes

Drug-loaded microspheres were clearly observed on the surface of the membrane compared with smooth, untreated PET stents (Figure 5A). Electrostatic adsorption was used to attach microspheres. Increased numbers of negatively charged microspheres could be absorbed by the positively charged film with increased dipping times. The dense microspheres fixed in the voids of the stents prevented endoleaks and retained the mechanical properties of the stents.

Excellent water permeability is an essential requirement for vascular stent-grafts, which is related to the thickness and morphology of the coated membrane. Generally, the water permeability should not exceed 300 mL/(cm^2^·min) to ensure the non-bleeding performance of artificial blood vessels [41]. As shown in Figure 5B, the thickness of the drug-loaded membranes increased from 0.113 ± 0.006 mm to 0.121 ± 0.012 mm after seven times conducting microsphere coating (*p* > 0.05), indicating that the modification of the drug-loaded membrane does not affect the thickness of the stent-grafts. The water permeability of the drug-loaded stent-grafts dropped from 163.33 ± 15.28 mL/cm^2^·min to 158.33 ± 17.56 mL/cm^2^·min after the modification of microspheres (*p* > 0.05), which showed that the layer-by-layer self-assembly method of microspheres coating did not greatly affect the water permeability of the stent-grafts.

The good longitudinal tensile strength provided excellent extensibility of the stent grafts and ensured that there was no major constriction of the stent graft when it opened during surgery by an airbag or by self-expansion [42]. Good radial tensile strength prevented the stent graft from breaking or slipping due to the excessive radial support force of the threads [43]. After seven cycles of microsphere coatings, the longitudinal tensile strength of the stent-graft increased from 81.36 ± 1.89 MPa to 82.58 ± 3.03 MPa. The radial tensile strength slightly decreased to 6.12 ± 0.88 MPa (Figure 5C), showing that the electrostatic adsorption of microspheres does not affect the macro-mechanical properties of the membranes (*p* > 0.05). Since the physical method of attracting positive and negative charges between the microspheres and the coating membrane was used, the microspheres were adsorbed on the surface of the stent-graft without affecting the internal structure of the membrane so the mechanical properties of the coating film could remain stable.

The good surface hydrophilicity of stent-grafts can help cell growth and coagulation. The water contact angle (WCA) of the uncoated membrane is 112.73 ± 2.04° at the beginning of the measurement, and the WCA decreased dramatically in 20 s, dropping to 2.07 ± 0.87° (Figure 5D). By contrast, the WCA of seven-cycle drug-loaded stent-grafts remained at 117.63 ± 3.33° after 20 s. The drug-loaded stent grafts exhibited improved hydrophobicity because PTX was hydrophobic and dissolved poorly in water. Additionally, PEG and MT promoted the transformation of the secondary structure of SF from random coils to β-sheets, which increased the water insolubility of SF microspheres [44,45,46].

### 3.6. Structural Analysis and Drug Release of Stent-Grafts with Drug-Loaded Membranes

As shown in Figure 5E, the stent-grafts with drug-loaded membranes had absorption peaks at 2960 cm^−1^, which mainly corresponded to the stretching vibration peaks of -CH_3_ in PTX and MET, indicating that the functional microspheres were attached to the surface of stent-grafts. The absorption peak intensity at 1098 cm^−1^ and 1247 cm^−1^ of the stent-grafts was significantly increased compared with the untreated stents, corresponding to the stretching vibration peak of the C-O group bond and the stretching vibration peak of the O-C-O bond, respectively, which further confirmed that the drug-loaded SF microspheres were coated on the surface of stent-grafts.

The in vitro release of stent-grafts with drug-loaded membranes was determined. The drug-loaded membranes exhibited a sudden release within the first 8 h and kept a slow cumulative release rate, which was reached between 8 and 72 h. The cumulative release of MET and PTX reached 51.3% and 21.09%, respectively (Figure 5F). This indicates that the remaining drugs were released slowly as the SF microspheres decomposed, resulting in long-term drug delivery.

### 3.7. Cell Inhibition of Stent-Graft with Drug-Loaded Membranes

The growth inhibitory effect of the drug-loaded membrane on HUVECs was verified. In addition, the effect of a combined medication was tested. As shown in Figure 6A(a), the cell inhibition rate of the drug-loaded membranes decreased over time. The growth inhibition rate of endothelial cells on PTX-loaded membranes presented a significant downward trend at 48 h compared with that of 24 h (*p* < 0.05). Notably, there was a significant difference in the inhibition of cell growth between PTX and PTX/MET-loaded membranes at 12 h (*p* < 0.05), suggesting that the addition of MET had a significant inhibitory effect on endothelial cell formation. The dual drug-loaded membranes showed the best cell inhibition rate compared with single MET or PTX-loaded membranes, which were maintained for 60.51 ± 0.05% at 48 h. The characterization of cell viability could also reflect the cell inhibition rate. PTX/MET-loaded membranes showed a significant decrease compared to PTX-loaded membranes at 12 h (*p* < 0.05) and further confirmed that the addition of dual drugs had an inhibitory ability on cell growth. In contrast to the cell inhibition rate, the cell viability of drug-loaded membranes increased as the cell proliferation time increased (Figure 6A(b)). For example, the cell viability of PTX-loaded membranes increased significantly from 24 h to 48 h (*p* < 0.05). However, the cell viability of PTX/MET-loaded membranes decreased significantly during this period. The cell viability of PTX/MET-loaded membranes reached the lowest 43.72 ± 0.022% at 48 h compared to single drug-loaded membranes, exhibiting the best cytostatic effect consistent with experiments of cell inhibition.

The HUVECs in the blank group grew in multiple layers and overlapped each other, with clear borders and strong cell viability (Figure 6B(a_i_)). With the addition of drugs in the membranes, the growth status of endothelial cells in groups of PTX and MET-loaded membranes showed lower cell density and weakened fluorescence intensity. The endothelial cell density of group (Figure 6B(d_i_)) showed a significant downward trend, indicating that the PTX/MET-loaded membranes had a strong inhibitory effect on the growth of endothelial cells, consistent with the expected test results. As anti-tumor drugs, PTX and MET play a synergistic role in inhibiting the growth of tumor cells, and the experimental results of the drug-loaded membrane covering inhibiting the growth of endothelial cells provide a theoretical basis for further clinical application of artificial vascular stent-grafts.

## 4. Conclusions

Conventional vascular stent grafts for the treatment of AAA often lack the capability of sustained cell growth inhibition. In this work, woven vascular stent-grafts with SF-PTX-MET-MT microspheres modification were developed to inhibit cell growth. This was tested with HUVECs in vitro. Particularly, SF-PTX-MET-MT microspheres were synthesized by emulsification-precipitation technology and layer-by-layer coated on the surface of a woven stent with electrostatic bonding. SF was used as a drug carrier, and the high hydrophilicity of PEG and MT was used to induce the changes in SF molecular structures, resulting in the self-assembly of microspheres. The addition of PTX and MET made the microspheres porous and small. This small structure could increase the specific surface area and promote the release and dissolution of drugs. Dual drug coatings showed good sustained-release phenomena and growth-inhibitory properties of HUVECs. Stable release kinetics for more than 70 h and the cumulative release of MET and PTX reached 51.3%, and 21.09% were observed. We expected the remaining active ingredients to be released slowly as the SF microspheres decomposed, resulting in long-term drug delivery. Therefore, these woven vascular stent grafts with SF-PTX-MET-MT microspheres have the potential to inhibit thrombosis and the growth of AAA, which is expected to lead to the efficient treatment of the diseased tumor and improve clinical applications.

## Figures and Tables

**Figure 1 bioengineering-10-00399-f001:**
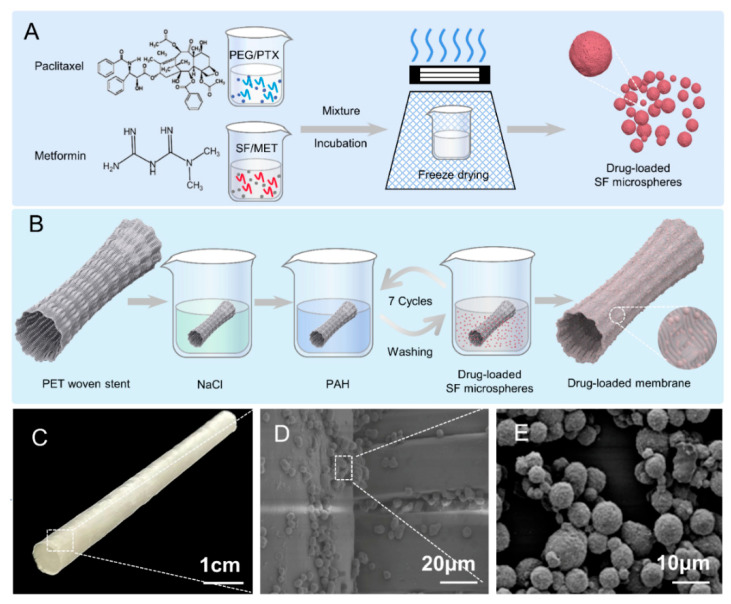
Schematic illustration of preparation and evaluation of stent-graft fabrics. (**A**) Preparation flow chart of drug-loaded SF microspheres. (**B**) Preparation flow chart of drug-loaded SF microspheres stent-graft. (**C**) Photography of woven vascular stent-grafts modified with drug-loaded SF microspheres. (**D**) SEM image of drug-loaded SF microspheres coated on the woven vascular stent. (**E**) SEM image of drug-loaded SF microspheres.

**Figure 2 bioengineering-10-00399-f002:**
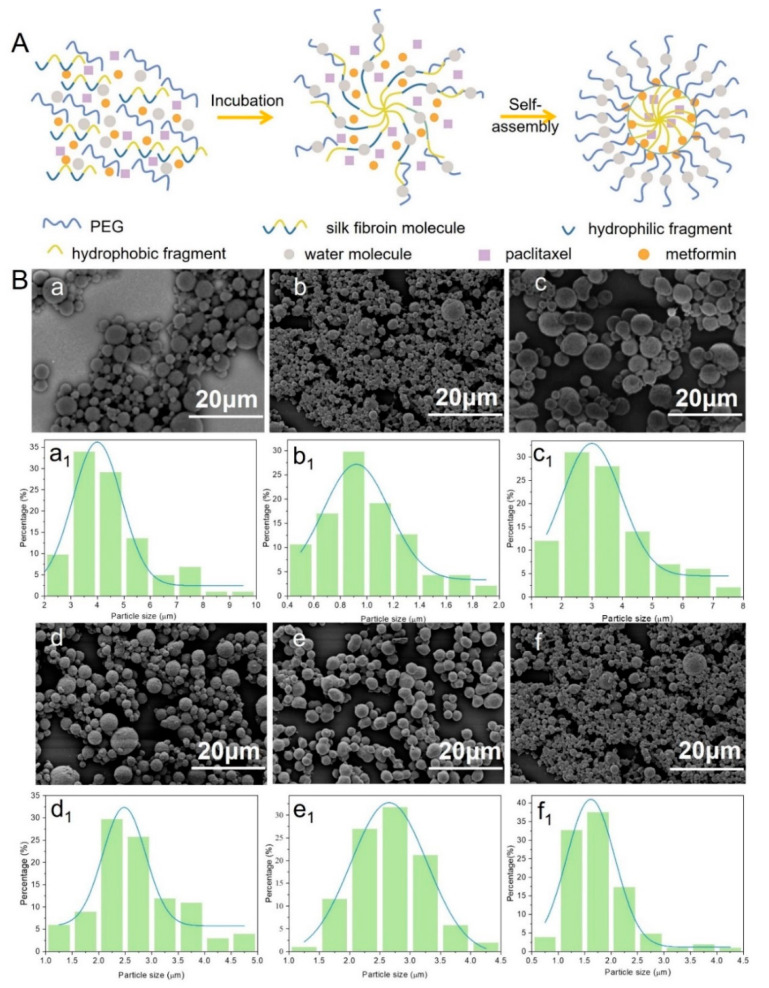
(**A**) Pellet-forming mechanism of drug-loaded silk fibroin microspheres; (**B**) Electron microscopy (**a**–**f**) and particle size distribution (**a_1_**–**f_1_**) of drug-loaded microspheres prepared with different parameters: (**a**,**a_1_**) SF, (**b**,**b_1_**) SF-PTX, (**c**,**c_1_**) SF-MT, (**d**,**d_1_**) SF-PTX-MT, (**e**,**e_1_**) SF-MET-MT, (**f**,**f_1_**) SF-PTX-MET-MT.

**Figure 3 bioengineering-10-00399-f003:**
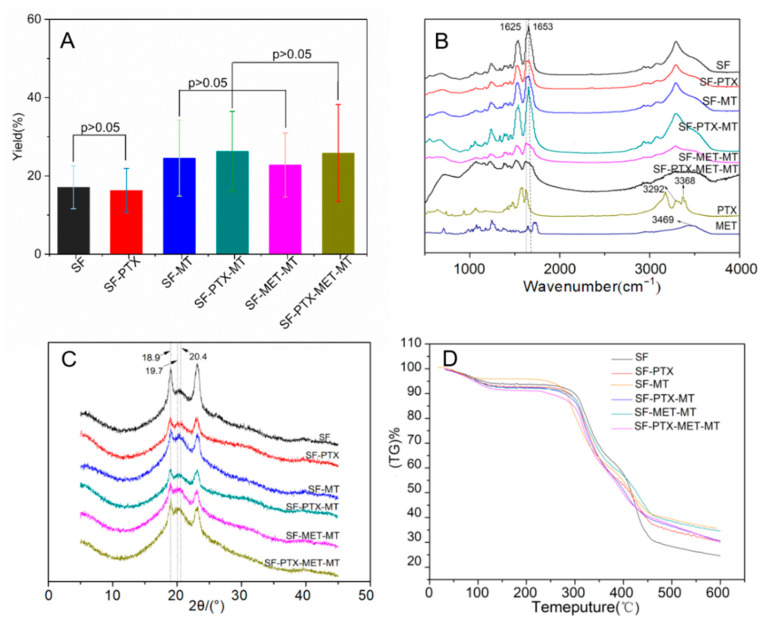
Yield analysis and the secondary structure of SF microspheres prepared with different parameters: (**A**) Yield distribution; (**B**) FTIR image; (**C**) XRD spectrum; (**D**) TGA image.

**Figure 4 bioengineering-10-00399-f004:**
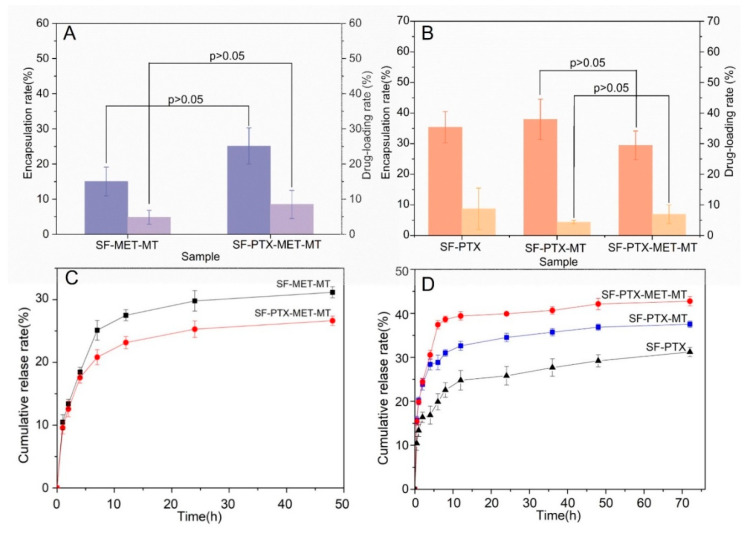
Quantitative characterization and cumulative release rate of drug-loaded SF microspheres prepared with different parameters: (**A**) MET; (**B**) PTX; (**C**) MET in vitro sustained release curve; (**D**) PTX in vitro sustained release curve.

**Figure 5 bioengineering-10-00399-f005:**
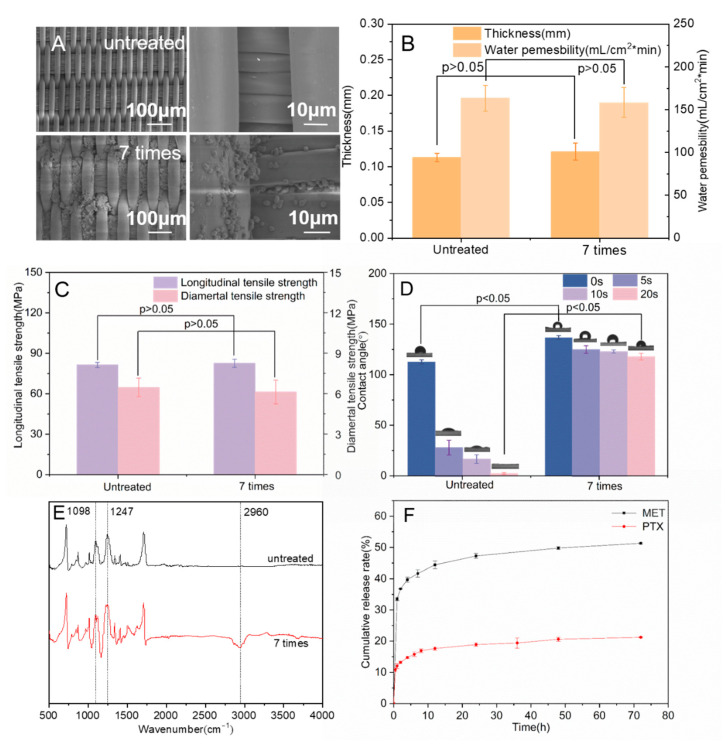
(**A**) SEM images of stent-graft modified with/without drug-loaded SF microspheres; (**B**) The thickness and water permeability of the stent-graft; (**C**) The longitudinal tensile strength and radial tensile strength of the stent-graft; (**D**) Water contact angle of the stent-graft; (**E**) FTIR images of the stent-graft; (**F**) Drug release curve in vitro of the stent-graft.

**Figure 6 bioengineering-10-00399-f006:**
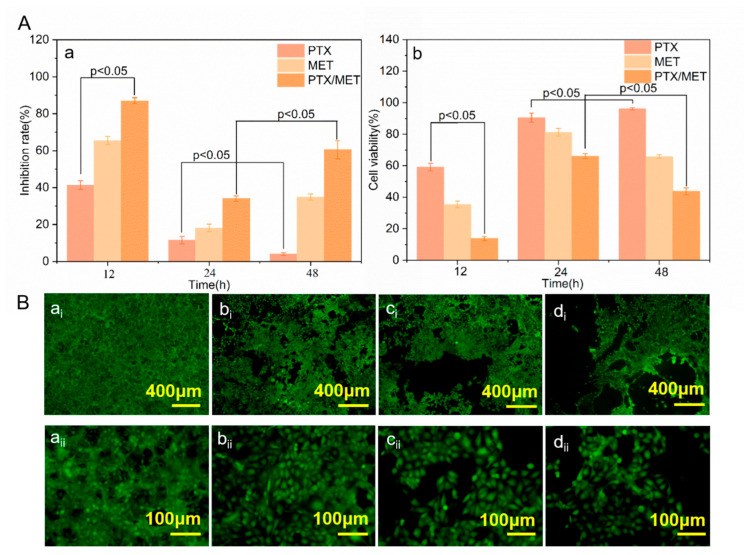
(**A**) Inhibition rate and cell viability of HUVECs: (**a**) Inhibition rate (**b**) Cell viability; (**B**) Growth status of HUVECs at 48 h: (**a_i_**,**a_ii_**) Cell growth between drug free stent samples; (**b_i_**,**b_ii_**) Cell growth between PTX coated stent samples; (**c_i_**,**c_ii_**) Cell growth between MET coated stent samples; (**d_i_**,**d_ii_**) Cell growth between PTX/MET coated stent samples.

## Data Availability

Data are available within the article.

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
