# Peer review of "Woven Vascular Stent-Grafts with Surface Modification of Silk Fibroin-Based Paclitaxel/Metformin Microspheres"

_bioengineering, 2023, doi:10.3390/bioengineering10040399_

Round 1

Reviewer 1 Report

This manuscript reports a Woven vascular stent-grafts with surface modification of silk fibroin-based paclitaxel/metformin microspheres. The study is well done, and claims are supported by the provided data. However, introduction part should be revised by indicating the efficacy of with surface modification of silk fibroin-based paclitaxel/metformin microspheres with comparing other reported work and authors need to include recent references. The manuscript could be acceptable for publication after revised the introduction part. 

Reviewer 2 Report

I have reviewed the paper entitled "Woven vascular stent-graft with SF-PTX-MET-MT microspheres for inhibiting AAA growth" for review. After carefully reading the manuscript, I believe that your study presents an innovative approach for the treatment of AAA.

However, I recommend some modifications in the Introduction section to provide more context for the readers. Specifically, it would be helpful to briefly discuss the current challenges in the treatment of AAA and the limitations of conventional vascular stent grafts. Also, MgO-Silk based composites should also be introduced as an innovative method for the treatment of the cancer cells in the introduction ex Nanomaterials 11 (3), 695. Additionally, it would be beneficial to provide some background information on woven vascular stent-grafts and their potential advantages over conventional stent grafts.

Furthermore, I suggest that you expand on the results section to provide more details on the experiments and findings. Specifically, it would be helpful to describe the synthesis and characterization of SF-PTX-MET-MT microspheres in more detail, including the methods used to determine the specific surface area and drug release profiles. Additionally, it would be beneficial to provide more information on the in vitro testing of the stent-grafts, including the methods used to assess growth inhibition and the results obtained.

The manuscript can be accepted with minor revision.

Reviewer 3 Report

1. Images in current figure 1C must be renumbered as parts c,d, and e to avoid confusion.

2.Vascular stent grafts typically would encounter 3D cellular microenvironment and not necessarily a monolayer, why did the authors choose a monolayer cell culture model for their cell study?

3.Cell viability for the PTX/MET and  MET groups seems to be decreasing significantly over a very short period of time 48 hours. This is not helpful in physiological scenarios where these vascular stents need to be functional for longer time periods. What is the proposed advantage of these two groups if the viability cannot be sustained or improved over 48-hour cultures. How does this data compare to control groups? It is important to statistically compare and discuss viability trends within each group across different time points. Please update the viability plot accordingly.
